# Efficacy and Safety of Cytisine in Combination with a Community Pharmacists’ Counselling for Smoking Cessation in Thailand: A Randomized Double-Blinded Placebo-Controlled Trial

**DOI:** 10.3390/ijerph192013358

**Published:** 2022-10-16

**Authors:** Pum Phusahat, Piyameth Dilokthornsakul, Watchara Boonsawat, Uraiwan Zaeoue, Nadthatida Hansuri, Nirachara Tawinkan, Ampornpan Theeranut, Sunee Lertsinudom

**Affiliations:** 1Division of Pharmacy, Somdej Phra Yupparat Sa Kaeo Hospital, Sa Kaeo 27000, Thailand; 2Center for Medical and Health Technology Assessment (CM-HTA), Department of Pharmaceutical Care, Faculty of Pharmacy, Chiang Mai University, Chiang Mai 50200, Thailand; 3Department of Pharmacy Practice, Faculty of Pharmaceutical Sciences, Naresuan University, Phitsanulok 65000, Thailand; 4Srinagarind Hospital, Faculty of Medicine, Khon Kaen University, Khon Kaen 40002, Thailand; 5Community Pharmacy, Faculty of Pharmaceutical Sciences, Khon Kaen University, Khon Kaen 40002, Thailand; 6Faculty of Nursing, Khon Kaen University, Khon Kaen 40002, Thailand; 7Division of Clinical Pharmacy, Faculty of Pharmaceutical Sciences, Khon Kaen University, Khon Kaen 40002, Thailand

**Keywords:** cytisine, efficacy, safety, smoking cessation, community pharmacist

## Abstract

Background: Cytisine is a prospective pharmacological alternative for community pharmacy smoking cessation services. However, it has not yet been licensed because of a lack of efficacy and safety information in Thailand. Objective: The aim of this study was to evaluate the efficacy of cytisine in combination with community pharmacists’ counselling on smoking cessation in a community pharmacy in ThailandDesign. Setting, participants, and interventions: A double-blinded randomized placebo-controlled trials was carried out. Participants aged >18 years old who smoked >10 tobaccos/day were randomly assigned to receive cytisine or placebo and five sessions of counselling by a community pharmacist. The primary outcome was a continuous abstinence rate (CAR) at week 48. The CAR was also measured at weeks 2, 4, 12, and 24. Adverse events were monitored. Results: A total of 132 participants were included, with 67 receiving cytisine and 65 receiving a placebo. Approximately 95% of participants were male. The CARs were determined to be 14.93% and 6.15% for cytisine and placebo, respectively, at week 48. The relative risk (RR) was 2.41 (95% confidence interval (CI); 0.80–7.35, *p* = 0.102). The RRs for CAR at weeks 2, 4, 12, and 24 were 2.43, 2.91, 2.50, and 1.78, respectively. Only the RRs for weeks 2, 4, and 12 were statistically significant. Common and non-serious gastrointestinal and neurological adverse events were observed. Conclusion: Cytisine, when combined with community pharmacists’ counselling, did not statistically improve the CAR at week 48, although it did improve the CAR at weeks 2, 4, and 12. Adverse events of cytisine were common and non-serious (registration number: TCTR20180312001).

## 1. Introduction

Tobacco smoking is one of the significant health problems worldwide. It causes several non-communicable diseases, such as chronic obstructive pulmonary disease, emphysema, cardiovascular disease, and cancer. It is one of the leading causes of premature death [1,2]. Approximately 1.3 billion people use tobacco products, 80% of whom are in low-and-middle income countries. In addition, tobacco smoking causes approximately 8 million deaths/year [2].

Pharmacotherapy has been proven to be an effective alternative for tobacco smoking cessation, especially for heavily nicotine-dependent smokers. Several studies indicate that smokers with pharmacotherapy have a 10–35% higher opportunity to quit smoking than smokers without pharmacotherapy. Moreover, smokers with pharmacotherapy and behavioral and psychosocial support have a 30–40% higher opportunity to quit smoking [3]. Pharmacotherapy could reduce smoking withdrawal symptoms. To date, two groups of medications have been approved for smoking cessation, including nicotine-replacement therapy (NRT) and non-NRT. When compared to non-pharmacotherapy, the continuous abstinence rate (CAR) increased by 1.5–1.9 folds. Bupropion hydrochloride and varenicline, the non-NRT medications, show a higher opportunity to quit smoking by 2.0 and 3.1 folds, respectively. Several medications have also been investigated as options for smoking cessation, such as *Vernonia Cinerea* [4] and cytisine [5,6,7]. 

Cytisine, a natural compound extracted from *Laburnum anagyroides*, is structurally similar to nicotine. It can bind to the alpha-4 beta-2 nicotinic acetylcholine receptor (nAChR) subtype with high affinity. The binding results in reduced nicotine withdrawal symptoms through the partial simulation of nAChR, which can moderately release dopamine and block the effect of nicotine from tobacco on the receptor [8,9]. A previous meta-analysis indicates that cytisine is more effective than placebo for a >6 months abstinence rate [10]. In addition, it seems to be more effective than varenicline and has a lower total cost [11]. However, an updated head-to-head randomized controlled trial shows the failure to be non-inferior to cytisine, compared to varenicline [12]. 

In Thailand, smoking cessation services have been implemented in various settings, such as hospitals and community pharmacies. Cytisine is a potential pharmacotherapy option for smoking cessation, along with community pharmacists’ counselling. However, cytisine has not yet been approved. Evidence of the efficacy of cytisine in combination with community pharmacists’ counselling in Thailand is still required. Therefore, this study aims to determine the efficacy and safety of cytisine as a pharmacotherapy option, in combination with community pharmacists’ counselling for smoking cessation, compared to community pharmacists’ counselling with placebo, in the context of community pharmacies in Thailand. 

## 2. Materials and Methods

### 2.1. Study Design and Participant Selection

This study was of a randomized single-center, double-blind, parallel design. It was conducted in a pharmacy at the Faculty of Pharmaceutical Sciences, Khon Kaen University. It was approved by the Khon Kaen University Ethics Committee in Human Research, according to the Declaration of Helsinki and Good Clinical Practice guidelines (ICH GCP), No. HE601321. All participants were informed of the study information and provided informed consent. This study protocol was registered at www.thaiclinicaltrials.org (accessed on 15 January 2022) (TCTR20180312001). 

Participants who were aged of 18–65 years old, smoked >10 cigarettes/day, were willing to quit smoking at a preparation level based on the trans-theoretical model [13], and could be contacted by a phone call were eligible. Female participants of reproductive age were required to have a contraception before and during the study. Participants with the following clinical conditions were excluded; cardiac arrhythmia, cardiovascular disease, cancer, chronic renal disease (eGFR ≤ 30 mL/min/1.73 m^2^), psychiatric disorders (depression, schizophrenia, or using other drugs, such as marijuana and amphetamines), being pregnant, breast-feeding, or being treated with other types of smoking cessation medications. The study was conducted from June 2018–March 2021.

### 2.2. Study Intervention

Participants were randomly assigned to the either active drug or placebo in a 1:1 ratio. Participants in the active drug group received film-coated cytisine tablets, and participants in the placebo group received matching film-coated placebo tablets. Cytisine treatment regimen was as follows: six tablets of 1.5 mg per day for days 1–3 (one tablet every 2 h), five tablets per day for days 4–12 (one tablet every 2.5 h), four tablets per day for days 13–16 (one tablet every 3 h), three tablets per day for days 17–20 (one tablet every 5 h), and two tablets per day for days 21–25 (one tablet every 6 h). The dosing regimen was based on a previous study in eastern Europe. In addition, the regimen was approved by Health Canada as a natural health product [14,15]. In addition to study medication, all participants received smoking cessation counselling by trained pharmacists at a community pharmacy a total of five times (before participation, week 1, week 2, week 4, and week 12). Two trained pharmacists were randomly selected to provide smoking cessation counselling and followed-up with participants in this study. Each participant met with the same pharmacist throughout the counselling and follow-up periods. The 5As model—ask, advise, assess, assist, and arrange—was used as a counselling guide to help participants quit smoking [16]. Briefly, participants were asked their health profiles and smoking history and advised to quit smoking. If participants decided to participate and provide informed consent, pharmacists assessed patients’ willingness to quit smoking using the trans-theoretical model [13] and nicotine dependence using the Fagerström test for nicotine dependence score (FTND) [17]. After the completion of the assessment, pharmacists provided patient education on the use of medication, health behavior modification, and other smoking- and smoking cessation-related information and collected baseline information. Finally, pharmacists arranged for the smoking cessation start date and follow-up visits. Cytisine tablets and matching placebo were prepared and provided by the Government Pharmaceutical Organization, the Thai government pharmaceutical manufacturer. Both participants and care providers were blinded. The manufacturer had no important role in this study, except providing medications.

### 2.3. Study Procedure

Stratified randomization was performed at the setting. The FTND was used as the stratifying factor, as 0–3 (minimally nicotine-dependent) and 4–10 (moderately-to-highly nicotine-dependent) [17], because it was associated to successful smoking cessation rate. A lottery ticket method was used for random participants to receive either cytisine or a placebo. Lottery tickets were prepared and numbered. Ten tickets were put in each of two opaque containers. One was for participants with minimal nicotine dependence, and the other one was for participants with moderate-to-high nicotine dependence. The containers were re-filled when all ten tickets were drawn. The procedure was repeated until we reached the estimated number of participants. The procedures and randomization results were concealed and blinded to both providers and participants.

### 2.4. Outcomes and Follow-Up

The primary outcome was self-reported continuous abstinence (CA) at week 48, confirmed by exhaled carbon monoxide (exhaled CO) <7 ppm. The CAR was also assessed at weeks 2, 4, 12, and 24. Secondary outcomes were 7-day self-reported point prevalence abstinence (PA), exhaled CO, relapse, health-related quality of life measured by WHOQOL-BREF-THAI and EQ-5D-5L (Thai version), and adverse events. The relapse rate was defined as the participant re-smoked >1 time after quitting smoking. Self-reported adverse events were observed for up to 4 weeks because the interventions were given to participants for only 25 days. In addition, laboratory monitoring was also performed at baseline and week 4, including serum creatinine (SCr), estimated glomerular infiltration rate (eGFR), alanine aminotransferase (ALT), electrocardiography (EKG), %peak expiratory flow rate (%PEFR), and forced expiratory volume at the 1st second (FEV_1_). The details of the participant’s follow-ups are presented in Figure 1.

### 2.5. Statistical Analyses

With a previous proportion of smoking cessation in patients using cytisine (0.40) [7] and placebo (0.15) [15], a total of 100 participants was required to detect the difference in CAR, with 80% power and an alpha level of 0.05. However, based on the estimated drop-out rate of 30%, the estimated total number of participants was 130.

Intention-to-treat analysis was applied, with the assumption that participants who lost to follow-up had failed to quit smoking. Chi-square was used to test the primary outcome and other dichotomous outcomes, while a *t*-test was used to test continuous outcomes. For each outcome, the risk ratio (RR) and mean difference (MD), as well as the corresponding 95% confidence interval (CI), were calculated. A subgroup analysis by FTND as minimally dependent (FTND 0–3), moderately dependent (FTND 4–6), and highly dependent (FTND 7–10) was performed.

## 3. Results

### 3.1. Participants’ Characteristics

A total of 132 participants were included. Sixty-seven participants were randomly assigned to receive cytisine, while 65 participants received placebo. However, only 29 participants (43.3%) for the cytisine group and 31 participants (47.7%) for the placebo group remained at week 48 (Figure 2).

The average age was 43.80 ± 11.76 years old for the cytisine group and 42.46 ± 14.59 years old for the placebo group. More than 90% of participants were male for both groups. The average duration of smoking was 24.06 ± 13.41 years for the cytisine group and 21.29 ± 13.84 years for the placebo group. The nicotine dependence score, measured by the FTND, was similar between groups (5.00 ± 2.35 Vs. 4.25 ± 2.05; *p* = 0.053). The details of the participants’ characteristics are presented in Table 1.

### 3.2. Continuous Abstinence Rate

A total of 10 participants (14.93%) in the cytisine group had CA at week 48, while 4 participants (6.15%) in the placebo group had the outcome. At week 48, the RR for CAR was 2.41 (95% confidence interval (CI); 0.80, 7.35; *p* = 0.10). In addition, the CARs at week 2, week 4, and week 12 were significantly higher in the cytisine group, but it was not significant at week 24. A subgroup analysis indicated that the significant difference in CAR was observed in moderately nicotine-dependent participants at week 2, week 4, and week 12. No difference was observed in minimally nicotine-dependent and highly nicotine-dependent participants (Table 2).

### 3.3. The 7-Day Point Prevalence Abstinence Rate

Sixteen participants (23.88%) in the cytisine group had 7-day PA at week 48, while 12 participants (18.46%) in the placebo group had 7-day PA. At week 48, the RR for PAR was 1.29 [95%CI; 0.66, 2.52; *p* = 0.45]. PARs at week 2 and week 4 in the cytisine group were significantly higher than those in the placebo group. However, no significant differences between the groups were observed at week 12 and week 24 (Table 3).

### 3.4. Exhaled Carbon Monoxide

At baseline, there was no statistically significant difference in exhaled CO between groups (13.91 8.07 vs. 13.15 9.75; *p* = 0.628). At week 2 (MD = −2.18 (95%CI; −3.78, −0.58); *p* = 0.008), week 4 (MD =−1.60 (95%CI; −3.19, −0.02); *p* = 0.047), and week 12 (MD = −2.52 (95%CI; −4.63, −0.40); *p* = 0.020), the exhaled CO was statistically lower in the cytisine group. However, at weeks 24 (0.77 (95% CI; −2.38, 3.93); *p* = 0.625) and 48 (MD = 0.93 (95% CI; −5.11, 4.70); *p* = 0.934), there were no differences between groups (Table 3).

### 3.5. Relapse

Only 30 participants in the cytisine group and 12 participants in the placebo group were eligible for the assessment of relapse rate. Of those, 20 participants (66.7%) in the cytisine group and 8 participants (66.7%) in the placebo group were defined as relapsed. The number of days to relapse was also not different between the groups. The average number of days to relapse in the cytisine group was 123.89 ± 82.34 days, while that of the placebo group was 111.57 ± 54.88 days (*p* = 0.718).

### 3.6. Health-Related Quality of Life

Patients’ health-related quality of life, measured by WHOQOL-BREF-THAI, improved from baseline to week 12 for both groups. It slightly decreased at week 24 and week 48, but it was still higher than baseline. The differences in the health-related quality of life between groups were not statistically significant at each measured time (Table 3).

Similarly, the health-related quality of life measured by EQ-5D-5L improved from baseline to week 12 for both groups and slightly decreased in week 24 and week 48. However, it was still higher than the baseline. No statistical significance for the health-related quality of life was observed in each of the measured time points (Table 3).

### 3.7. Adverse Events

A total of 37 adverse events (55.22%) were observed in the cytisine group, while 26 adverse events (40.00%) were observed in the placebo group. Most adverse events were common and non-serious. Most adverse events were diarrhea, abdominal distension, dizziness, drowsiness, dry mouth, and sore throat. The rates of each adverse event were similar between groups, except for headaches and insomnia. Three participants (4.48%) in the cytisine groups experienced headaches, but no participant in the placebo group had headaches. Similarly, eight participants (11.94%) in the cytisine group experienced insomnia, while only one participant in the placebo group experienced insomnia.

Additionally, the laboratory monitoring indicated the safety of cytisine. The average SCr of participants in the cytisine group at baseline was similar to that at week 4 (0.96 ± 0.19 g/dL vs. 0.95 ± 0.17 g/dL). In the placebo group, the average SCr at baseline and at week 4 were 0.98 ± 0.24 g/dL and 0.96 ± 0.28 g/dL, respectively. There was no statistical significance observed between groups (*p* > 0.05).

The average eGFR of the participants in the cytisine group was also similar to that of week 4 (99.40 ± 14.75 Vs. 98.55 ± 16.33 L/min). The average eGFR in the placebo group were 101.52 ± 22.43 L/min and 101.46 ± 28.54 L/min, respectively. There was no statistical significance observed between groups (*p* > 0.05).

The average ALT levels of participants in the cytisine group were 33.23 ± 25.58 g/dL at baseline and 43.10 ± 37.99 g/dL at week 4, while those of participants in the placebo group were 34.00 ± 25.75 g/dL at baseline and 33.04 ± 19.36 g/dL at week 4. There was no statistical significance observed between groups (*p* > 0.05).

The number of participants with an abnormal EKG was observed in both groups. At baseline, 19 of 56 participants (33.93%) who underwent EKG monitoring in the cytisine group had an abnormal EKG, while 22 of 57 participants (38.60%) in the placebo group had an abnormal EKG. There were no statistically significant differences (*p* = 0.606). At week 4, 10 of 48 participants (20.83%) in the cytisine group had an abnormal EKG, while 13 of 46 participants in the placebo group had an abnormal EKG. Moreover, the difference was also not statistically significant (*p* = 0.475).

The difference in %PEFR between the cytisine and placebo groups at baseline was not statistically significant (93.68% 15.45% vs. 93.78% 17.14%; *p* = 0.970). At week 4, the difference in %PEFR between the cytisine and placebo groups was also not statistically significant (114.00% 75.26% vs. 95.60% 21.50%; *p* = 0.095). The average FEV1 at the baseline of the participants in the cytisine group was 93.98 ± 11.22 L/min, while that of the participants in the placebo group was 93.94 ± 13.13 L/min. Moreover, the difference was not statistically significant (*p* = 0.986). At week 4, the difference in FEV1 between groups was also not statistically significant (95.66 ± 10.76 L/min VS. 92.07 ± 19.48 L/min; *p* = 0.269). All adverse events are presented in Table 4.

## 4. Discussion

This study was the first study in Thailand to assess the efficacy of cytisine in combination with smoking cessation counselling by a community pharmacist in a community pharmacy setting. We found that the CAR at week 48 in patients receiving cytisine with community pharmacists’ counselling was not statistically different from that in patients receiving community pharmacists’ counselling alone. However, cytisine tends to improve the smoking cessation rate in this setting, as it showed better CAR at weeks 2, 4, and 12. Cytisine had gastrointestinal adverse events, headaches, and insomnia. However, those adverse events were common and non-serious.

Even though we found no significant difference in CAR at week 48 between groups, the CAR in the cytisine group (14.93%) was 2.4-fold higher than that of the placebo group (6.15%). The effect size was lower than what was observed in a previous randomized controlled trial, which showed a 3.4-fold higher treatment effect size [15]. The difference in the observed effect size might be due to the smoking cessation counselling. In our study, trained community pharmacists provided smoking cessation counselling five times, which could help participants successfully quit smoking, whereas the previous study provided smoking cessation counselling once at baseline. The CAR at week 48 of the placebo group in the current study (6.15%) was approximately 2.5-fold higher than what was observed in the previous study (2.40%). Thus, the effect size observed in this study was possibly added to the effect of the community pharmacist’s counselling.

We also observed that both CAR and PAR gradually declined after week 4 of treatment initiation for both groups. It is common for smokers to re-use tobacco after the discontinuation of medications. The regimen of cytisine was only for 25 days. Participants might perceive that they discontinued smoking cessation because of no medication used and could not resist reusing tobacco.

Even though cytisine was likely to help people quit smoking, especially within 24 weeks after cytisine initiation, it had no effect on an individual’s health-related quality of life. The WHOQOL-BREF-THAI and EQ-5D-5L showed a similar trend, that the health-related quality of life of participants receiving cytisine was similar to those receiving a placebo. In addition, the health-related quality of life of participants in both groups was minimally better than baseline. Thus, past smokers who just finished a smoking cessation program provided by community pharmacists in Thailand might not have the feelings of a better quality of life and need motivation to continue stopping smoking.

One interesting adverse event of cytisine observed in this study was neurologic adverse events. Patients receiving cytisine had a higher number of neurologic adverse events than patients receiving a placebo. More than 10% of patients receiving cytisine experienced insomnia, while only 1.5% of patients receiving a placebo experienced insomnia. In addition, approximately 4.5% of patients receiving cytisine experienced headaches, but no patients receiving the placebo experienced headaches. Thus, community pharmacists or other healthcare providers who assist smokers to quit smoking should closely monitor the neurologic adverse events in patients initiating cytisine.

Other common and non-serious adverse events observed were consistent with those observed in previous studies [15,18]. They were gastrointestinal adverse events, including abdominal distension, dry mouth, sore throat, and constipation. The number of gastrointestinal adverse events was similar between the cytisine and placebo, which was <5%. However, community pharmacists or other healthcare providers should still provide the gastrointestinal adverse events to smokers who attempt to quit smoking using cytisine and monitor the adverse events.

Our findings could be used as important information for the Thai Food and Drug Administration to consider, regarding whether cytisine should be approved for smoking cessation in Thailand, based on the efficacy and safety observed in the Thai context. In addition, other healthcare policy makers, such as the National List of Essential Medicine (NLEM) committee, which is responsible for listing essential medications into NLEM and allows all people to access the medication if it is approved, could be used this information to consider whether cytisine should be added in NLEW.

The limitations of this study should also be discussed. First, we encountered a number of patients with loss to follow-up. We assumed that patients who lost to follow-up failed to quit smoking. This is subject to attrition bias. However, we tried to minimize it by attempting to contact patients who did not visit the pharmacy at their appointment. Unfortunately, only half of the patients could finish the study as planned. Second, even though we attempted to standardize the smoking cessation counselling by pharmacists, there might have been some variations in the smoking cessation counselling among the pharmacists. That might also affect the smoking cessation rate. We tried to minimize the variation by conducting a training session to all pharmacists who participated in our study and randomly selected pharmacists for giving patients’ counselling in both groups. Third, even though we recruited participants without any restriction on sex, almost all participants (95.5%) were male. This indicated that male smokers are more likely to seek out smoking cessation consultation at a pharmacy than female smokers. These limitations lead to a limited generalizability of our findings for female smokers. Forth, because this study was community pharmacy-based, we could not perform the serum cotinine test to confirm smoking abstinence. We, instead, used the exhaled CO to confirm the smoking abstinence. It appears that exhaled CO could be a good biomarker for smoking and nicotine dependence. Our findings from exhaled CO were consistent with the self-reported CAR. The exhaled CO was significantly lower in participants receiving cytisine than those receiving placebo at weeks 2, week 4, and week 12.

Last, the study was conducted in a pharmacy at the faculty of pharmacy in a university. The effect of smoking cessation counselling might be different in other pharmacy setting, such as chain pharmacies or private stand-alone pharmacies. Thus, the observed effect of cytisine in such settings might be different from our setting. The generalizability of our findings might be interpreted with caution.

## 5. Conclusions

Cytisine, in combination with five sessions of community pharmacists’ counselling, could not statistically improve CAR at week 48, compared to five sessions of community pharmacists’ counselling alone. However, it tends to improve the CAR at weeks 2, 4, and 12. Cytisine had gastrointestinal adverse events, headaches, and insomnia. However, those adverse events were common and non-serious.

## Figures and Tables

**Figure 1 ijerph-19-13358-f001:**
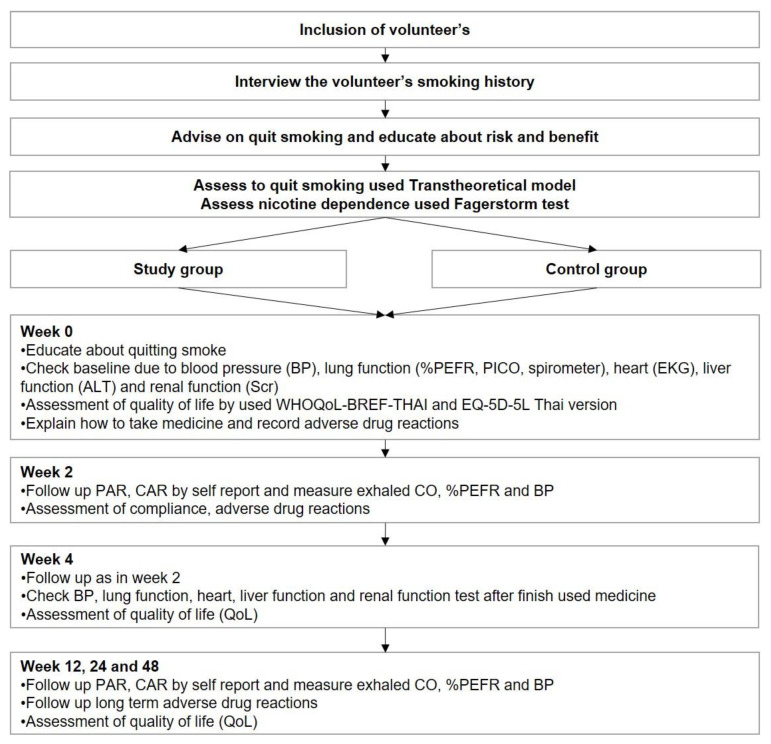
Participant inclusion and follow-up.

**Figure 2 ijerph-19-13358-f002:**
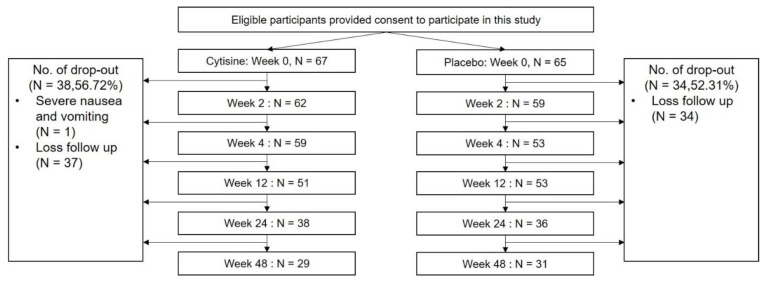
A Flow of participants.

**Table 1 ijerph-19-13358-t001:** Participants’ characteristics.

Participants’ Characteristics	Cytisine (*n* = 67)	Placebo (*n* = 65)	*p*-Value *
Male, number (%)	63 (94.03)	63 (96.92)	0.680
Age (years)	43.80 ± 11.76	42.46 ± 14.59	0.562
Body mass index kg/m^2^	23.76 ± 4.77	24.27 ± 4.18	0.515
Duration of smoking (years)	24.06 ± 13.41	21.29 ± 13.84	0.245
Number of cigarettes per day	19.04 ± 9.27	16.88 ± 5.35	0.105
The Fagerstöm test for nicotine dependence	5.00 ± 2.35	4.25 ± 2.05	0.053
Exhaled CO	13.91 ± 8.07	13.15 ± 9.75	0.626
FEV_1_ (L/min)	92.63 ± 10.96	93.32 ± 12.34	0.735
%PEFR	93.67 ± 15.45	93.78 ± 17.14	0.969
Systolic blood pressure (mmHg)	126.63 ± 13.82	127.46 ± 16.23	0.752
Diastolic blood pressure (mmHg)	79.36 ± 9.68	82.14 ± 12.68	0.158
Serum creatinine (g/dL)	0.94 ± 0.16	0.99 ± 0.22	0.137
eGFR (ml/min)	101.02 ± 16.98	100.45 ± 20.89	0.864
ALT (U/L)	40.44 ± 45.91	33.70 ± 24.98	0.299

**Abbreviations**: ALT, alanine transaminase; CO, carbon monoxide; eGFR, estimated glomerular filtration rat; FEV1, forced expiratory volume in the 1st second; g/dL, grams per deciliter; kg/m^2^, kilograms per square meter; L/min, liters per minute; ml/min, millimeters per minute; mmHg, millimeters of mercury; %PEFR, %peak respiratory flow rate; U/L, units per liter. * tested by independent *t*-test or Fisher’s exact test.

**Table 2 ijerph-19-13358-t002:** Result for continuous abstinence.

Outcomes	Cytisine (*n*, %) (*n* = 67)	Placebo (*n*, %) (*n* = 65)	Risk Ratio (95% CI)	*p*-Value *
**Overall participants**				
Week 2	30 (44.78)	12 (18.46)	2.43 (1.36, 4.31)	0.001
Week 4	30 (44.78)	10 (15.38)	2.91 (1.55, 5.46)	<0.001
Week 12	18 (26.87)	7 (10.77)	2.50 (1.12, 5.57)	0.018
Week 24	11 (16.42)	6 (9.23)	1.78 (0.70, 4.53)	0.219
Week 48	10 (14.93)	4 (6.15)	2.41 (0.80, 7.35)	0.102
**Participants with FTND ≤ 3 (minimally nicotine-dependent) (*n* = 17 vs. 20)**				
**Week 2**	6 (35.29)	4 (20.00)	1.76 (0.59, 5.23)	0.296
**Week 4**	6 (35.29)	3 (15.00)	2.35 (0.96, 8.01)	0.152
**Week 12**	5 (29.41)	2 (10.00)	2.94 (0.65, 13.27)	0.133
**Week 24**	5 (29.41)	2 (10.00)	2.94 (0.65, 13.27)	0.133
**Week 48**	5 (29.41)	2 (10.00)	2.94 (0.65, 13.27)	0.133
**Participants with FTND 4–6 (moderately nicotine-dependent) (*n* = 30 vs. 36)**				
**Week 2**	16 (53.33)	7 (19.44)	2.74 (1.30, 5.77)	0.004
**Week 4**	16 (53.33)	6 (16.67)	3.20 (1.43, 7.14)	0.002
**Week 12**	10 (33.33)	4 (11.11)	3.00 (1.05, 8.60)	0.028
**Week 24**	4 (13.33)	3 (8.33)	1.60 (0.38, 6.60)	0.513
**Week 48**	4 (13.33)	1 (2.78)	4.80 (0.57, 40.68)	0.107
**Participants with FTND ≥ 7 (highly nicotine-dependent) (*n* = 20 vs. 9)**				
**Week 2**	8 (40.00)	1 (11.11)	3.60 (0.53, 24.66)	0.120
**Week 4**	8 (40.00)	1 (11.11)	3.60 (0.53, 24.66)	0.120
**Week 12**	3 (15.00)	1 (11.11)	1.35 (0.16, 11.27)	0.778
**Week 24**	2 (10.00)	1 (11.11)	0.90 (0.09, 8.69)	0.928
**Week 48**	1 (5.00)	1 (11.11)	0.45 (0.03, 6.42)	0.548

* Tested by chi-square at α < 0.05.

**Table 3 ijerph-19-13358-t003:** Results for secondary efficacy outcomes.

Outcomes	Cytisine	Placebo	Treatment Effects (95% CI)	*p*-Value *
**7-day point prevalence abstinence (reported as risk ratio)**				
Week 2	38 (56.71)	19 (29.23)	1.94 (1.26, 3.00)	0.001
Week 4	44 (65.67)	24 (36.92)	1.78 (1.24, 2.55)	0.001
Week 12	31 (46.27)	21 (32.31)	1.43 (0.93, 2.22)	0.100
Week 24	17 (25.37)	17 (26.15)	0.97 (0.54, 1.73)	0.918
Week 48	16 (23.88)	12 (18.46)	1.29 (0.66, 2.52)	0.446
**Exhaled carbon monoxide (ppm) (reported as mean difference)**				
Baseline	13.91 ± 8.07	13.15 ± 9.75	0.76 (−2.32, 3.84)	0.628
Week 2	4.78 ± 4.38	6.96 ± 4.34	−2.18 (−3.77, −0.58)	0.008
Week 4	4.45 ± 3.62	6.06 ± 4.66	−1.61 (−3.18, −0.12)	0.047
Week 12	4.83 ± 4.58	7.34 ± 5.49	−2.51 (−4.63, −0.40)	0.020
Week 24	6.93 ± 5.87	6.16 ± 6.52	0.77 (−2.38, 3.93)	0.625
Week 48	7.57 ± 8.36	7.77 ± 7.55	0.20 (−5.11, 4.70)	0.934
**Health related quality of life measured by WHOQOL-BREF-THAI** **(reported as mean difference)**				
Baseline	3.36 ± 0.58	3.42 ± 0.70	0.04 (−0.04, 0.03)	0.280
Week 4	3.86 ± 0.71	3.73 ± 0.60	0.13 (−0.25, 0.51)	0.688
Week 12	3.86 ± 0.71	3.73 ± 0.77	0.13 (−0.30, 0.57)	0.475
Week 24	3.68 ± 0.77	3.50 ±0.64	0.18 (−0.23, 0.59)	0.247
Week 48	3.68 ± 0.77	3.65 ± 0.79	0.03 (−0.04, 0.04)	0.990
**Health related quality of life measured by EQ-5D-5L (reported as mean difference)**				
Baseline	78.95 ± 18.79	78.86 ± 14.22	−0.27 (−10.41, 9.86)	0.406
Week 4	89.45 ± 11.44	85.31 ± 9.95	4.14 (−2.39, 10.66)	0.299
Week 12	88.90 ± 12.44	85.90 ± 9.42	3.00 (−3.71, 9.71)	0.167
Week 24	84.18 ± 23.38	81.36 ± 14.32	2.82 (−8.97, 14.61)	0.175
Week 48	88.18 ± 11.68	86.18 ± 13.04	2.00 (−5.53, 9.53)	0.789

* Tested by independent *t*-test at α < 0.05.

**Table 4 ijerph-19-13358-t004:** The rate of adverse events between the cytisine group and the placebo group.

Adverse Events	Cytisine, No. (%) (*n* = 67)	Placebo, No. (%) (*n* = 65)
Diarrhea	2 (2.99)	1 (1.54)
Nausea and vomiting	0 (0)	1 (1.54)
Abdominal distension	3 (4.48)	6 (9.23)
Headache	3 (4.48)	0 (0)
Dizziness	4 (5.97)	3 (4.62)
Drowsiness	5 (7.46)	3 (4.62)
Insomnia	8 (11.94)	1 (1.54)
Depression	1 (1.49)	2 (3.08)
Dry mouth	3 (4.48)	6 (9.23)
Dry throat	1 (1.49)	0 (0)
Sore throat	3 (4.48)	3 (4.62)
Constipation	1 (1.49)	0 (0)
Runny nose	1 (1.49)	0 (0)
Tongue numbness	2 (2.99)	0 (0)
Total	37 (55.22)	26 (40.00)

## Data Availability

Data used in this study are available upon appropriate requests to the corresponding author.

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
