# Peer review of "Efficacy and Safety of Cytisine in Combination with a Community Pharmacists’ Counselling for Smoking Cessation in Thailand: A Randomized Double-Blinded Placebo-Controlled Trial"

_ijerph, 2022, doi:10.3390/ijerph192013358_

Round 1

Reviewer 1 Report

Thank you for conducting this study, it was very interesting! Please find my overall comments below:

Introduction  -  your introduction was short, sweet, and to the point and provided great background about your study. 

Methods

1. Was there a particular reason that female participant were required to have contraception?

2. How did you decide on that particular dosing scheme?

3. The section marked 2.3 Study Procedure is difficult to follow as it is difficult to understand what the purpose of this was. Perhaps you could elaborate a bit on this. 

4. I would be very interested in learning what the sessions with the community pharmacist entailed and the types of things that were discussed. Not necessary but could be something to consider adding. 

Results

1.  In your adverse events section, you write "most adverse events were un-common and serious" I understand what you are trying to say, but the way this sentence reads it makes it seem like the adverse events were serious but rare. 

2. Table 2 - Please moe it to the next page so that it does not get broken up between two pages

Reviewer 2 Report

General comments

This is an original monocentric double-blinded placebo-controlled trial which assesses the efficacy and safety of cytisine in combination with a community pharmacists' counselling for smoking cessation in Thai smokers.

The topic is of current interest as cytisine is largely used as a pharmacological aid for smoking cessation but studies on its use in the real-life are scarce.

The present study should be improved in the presentation of the methods and results, and also by expanding the discussion on the limitations.

Specific comments

Major

- More information on how the study subjects were selected should be provided in the Materials and Methods section. The limit of having studied a self-selected study population should be discussed. For example, the study subjects were mainly males. This should be discussed among the limitations of the study. The study took place over a period of almost 3 years. Thus, it seems that the study subjects were not consecutively enrolled. Are there information of the excluded subjects? 

- The study subjects were stratified for the level of nicotine dependence as assessed by the Fagerström test for nicotine dependence score. However the results of the study are presented considering the whole study population. Did a different efficacy of cytisine was observed in relation to different levels of nicotine dependence?

- Table 1. P values for the comparison between the Cytisine and placebo group should be provided. The placebo group shows a lower number of cigarettes smoked per day and a lower score of the Fagerström test for nicotine dependence. These differences should be discussed. Comparison analyses should take into account the differences between the two study groups.

- "...randomly select pharmacists for giving patients’ counseling in both groups". How many phamacists performed the counselling? Did the same pharmacists always perform counselling in the same study group? More detailed information should be provided in the Materials and Methods section.

- The lack of the measurement of the serum cotinine to confirm smoking abstinence should be discussed as a limitation of the study.

Minor

- Title "pharmacist’s counseling" should be "pharmacists' counselling".

- Page 6 of 13, section "3.3. The 7-day point prevalence abstinence rate". The description reported in this section refers to Table 3 (not to Table 2).

- Page 6 of 13, section "3.3. The 7-day point prevalence abstinence rate". "RR" should be "MD".

- Acronyms used in Fig. 1 should be expanded in the legend.

- Table 4. The number of the subjects in the Cytisine and placebo group should be reported.

- Table 1. "Nicotine dependence" should be the score of the Fagerstöm test for nicotine dependence.

- Substitute "Fagerstrom risk score" with "Fagerström test for nicotine dependence score".

- Substitute "Fagerstrom test" with "Fagerström test for nicotine dependence". Abbreviate "Fagerström test for nicotine dependence" as FTND.

- "the study was conducted in a pharmacy at the faculty of pharmacy in a university". This information should be reported in the Materials and methods section. The name of the University should be reported.

Round 2

Reviewer 2 Report

The authors have adequately revised the manuscript according to comments and it is now improved.

This reviewer has no further comments.